# Aging Favors Calcium Activation of Ryanodine Receptor Channels from Brain Cortices and Hippocampi and Hinders Learning and Memory in Male Rats

**DOI:** 10.3390/ijms26052101

**Published:** 2025-02-27

**Authors:** Jamileth More, José Pablo Finkelstein, José Luis Valdés, Cecilia Hidalgo, Ricardo Bull

**Affiliations:** 1Centro de Investigación Clínica Avanzada (CICA), Facultad de Medicina-Hospital Clínico, Universidad de Chile, Santiago 8380453, Chile; jamilethmore@gmail.com; 2Physiology and Biophysics Program, Faculty of Medicine, Institute of Biomedical Sciences, Universidad de Chile, Santiago 8380453, Chile; jfinkeht@gmail.com; 3Faculty of Medicine, Biomedical Neuroscience Institute, Universidad de Chile, Santiago 8380453, Chile; jlvaldes@uchile.cl; 4Department of Neuroscience, Faculty of Medicine, Universidad de Chile, Santiago 8380453, Chile; chidalgo@uchile.cl

**Keywords:** spatial memory training, aging, calcium signaling, antioxidants, RyR oxidation, thiol modification

## Abstract

The response of ryanodine receptor (RyR) channels to increases in free cytoplasmic calcium concentration ([Ca^2+^]) is tuned by several mechanisms, including redox signaling. Three different responses to [Ca^2+^] have been described in RyR channels, low, moderate and high activity responses, which depend on the RyR channel protein oxidation state. Thus, reduced RyR channels display the low activity response, whereas partially oxidized channels display the moderate response and more oxidized channels, the high activity response. As described here, RyR channels from rat brain cortices or hippocampi displayed aged-related marked changes in the distribution of these channel responses; RyR channels from aged rats displayed reduced fraction of low activity channels and increased fraction of high activity channels, which would favor Ca^2+^-induced Ca^2+^ release. In addition, compared with young rats, aged rats displayed learning and memory defects, with lower hit rates when tested in the Oasis maze, a dry version of the Morris water maze. Previous oral administration of *N*-acetylcysteine for 3 weeks prevented both the age-dependent effects on RyR channel activation by [Ca^2+^], and the learning and memory defects. Based on these results, it is proposed that redox-sensitive neuronal RyR channels partake in the mechanism underlying the learning and memory disruptions displayed by aged rats.

## 1. Introduction

Aging is often accompanied by a decline in cognitive function, which entails significant redox imbalance towards increased oxidation in brain cells [1]. The ryanodine receptor (RyR) calcium channels partake in hippocampal functions, including synaptic plasticity, considered the cellular basis of learning and memory, and in learning and spatial memory processes [2]. The redox state of some of its many cysteine residues modulates RyR channel activation, enabling these channels to act as cellular redox sensors [3,4,5,6]. A previous study of the type-1 (RyR1) channel isoform identified specific cysteines out of the 100 cysteines present in each RyR1 subunit and characterized their different redox modifications: *S*-nitrosylation, *S*-glutathionylation and oxidation to disulfides [7]. Furthermore, increasing the free calcium ion concentration ([Ca^2+^]) to the micromolar range promotes *S*-nitrosylation and/or *S*-glutathionylation of sulfhydryl (SH) residues of skeletal or cardiac RyR channels, resulting in enhanced Ca^2+^-induced Ca^2+^ release [8,9,10,11,12]. Moreover, Ca^2+^ release mediated by hippocampal RyR2 channels, the most abundant RyR channel isoform expressed in hippocampal tissue [13], is required for spatial learning and memory tasks [14]. Of note, when compared to young rats, aged rats display increased oxidation levels of the RyR2 isoform [15], which by generating anomalous Ca^2+^ signals due to RyR over activation by Ca^2+^, may contribute to the well-known impairments in hippocampal synaptic plasticity and spatial memory that take place during aging [1,2,16].

In particular, RyR channel redox state determines the three types of response (low, moderate or high) to activation by [Ca^2+^] of single RyR channels from rat brain cortex endoplasmic reticulum (ER) [17,18,19,20,21]. Thus, the addition of SH modifying agents while recording RyR channel activity causes a single low activity channel to adopt sequentially and stepwise, first the moderate and then the high activity response [18].

Our working hypothesis is that age-induced redox imbalance in the brain toward increased oxidation increases neuronal RyR channel activation by [Ca^2+^]; the resulting abnormal increase in cytoplasmic [Ca^2+^] in neuronal cells would lead to defective performance in a spatial memory task. To test this hypothesis, the performance in the Oasis maze of young (3-month-old) and aged (22-month-old) male rats, some of them fed with the antioxidant agent *N*-acetylcysteine (NAC), was studied; subsequently the in vitro responses to cytoplasmic [Ca^2+^] of single RyR channels, from the ER isolated from their respective hippocampal tissues, was determined.

## 2. Results

### 2.1. Effects of Age on the Ca^2+^-Induced Responses of RyR Channels from Rat Brain Cortex

The effects of age on the frequency of obtaining the three different responses to cytoplasmic [Ca^2+^] of single RyR channels was first tested in channels isolated from rat brain cortices of male rats with different ages (Figure 1). The RyR channels obtained from the rat brain cortex at all three ages studied displayed only three different response patterns, and were readily classified into low, moderate or high activity channels. However, the frequency of emergence of these three types of response was significantly different. The RyR channels from the brain cortex of 3-month-old rats, showed most frequently the low activity response (70.4%), followed by moderate (21.1%) and high (8.5%) activity responses. In contrast, single RyR channels from 22-month-old rats, displayed with similar frequency the moderate (44.8%) and the low (41.8%) activity responses, followed by the high activity response (13.8%), respectively (see Figure 1).

Accordingly, the ages of 3 and 22 months were chosen to perform training experiments in the Oasis maze and for single hippocampal RyR channel analysis.

### 2.2. Effects of Age on the Responses to [Ca^2+^] of Single Hippocampal RyR Channels

Representative current recordings taken from individual experiments showing the three different responses to [Ca^2+^] of single RyR channels from rat hippocampi are illustrated in Figure 2, whereas the Ca^2+^-dependence of the fractional open time (P_o_) values are depicted in Figure 3. As illustrated in Figure 3, the three responses to [Ca^2+^] displayed by hippocampal RyR channels, from both young and aged rats, were analogous. The curve-fitting of the data obtained with single RyR channels with the same Ca^2+^ response, from either aged or young rats, yielded similar parameter values. Hence, the continuous lines in Figure 3 were obtained using all data displayed by RyR channels from both young and aged rats.

Of note, the apparent affinities for activation and inhibition by Ca^2+^ were different for each channel response (Table 1). Low activity channels displayed significantly higher K_a_ values than moderate activity channels, and moderate activity channels exhibited higher K_a_ values than high activity channels. Conversely, the K_i_ values for low activity channels were lower than those exhibited by moderate activity channels, whereas moderate activity channels displayed lower K_i_ values than high activity channels (Table 1, Figure 3).

Although the three types of responses to cytoplasmic [Ca^2+^] were undistinguishable for single RyR channels obtained from the hippocampi of young or aged rats, their frequencies of emergence were quite different (Figure 4, histograms labeled C). In young rats, hippocampal RyR channels displayed most frequently the low activity response (66.7%), followed by the moderate (31.6%) and the high (1.7%) activity responses, whereas channels from aged rats displayed with similar frequency the moderate (35.7%) and the high (35.7%) activity responses, followed by the low (28.6%) activity response. Therefore, as shown by RyR channels from rat brain cortices, aging induced a marked change in the emergence of the different channel responses: it reduced the fraction of low activity channels from 66.7% to 28.6% and increased the fraction of high activity channels from 1.7% to 35.7%.

### 2.3. Effects of Age on Spatial Learning and Memory, and Protective Effects of NAC Feeding

To test the effects of aging on spatial learning and memory, young and aged rats were trained and tested in the Oasis maze (see Material and Methods). First, the behavior of young and aged rats on a learning task was explored. To this aim, the position of the reward well (water) was changed in each session. As illustrated in Figure 5A, the hit rate, defined as the number of times the rat found the reward during daily sessions composed of ten trials, was significantly different between young and aged rats at all times tested, from 24 h to 48 h and 72 h. Thus, young rats always found the reward in each of the three test sessions, with a hit rate value = 1. In contrast, aged rats displayed significant defects in learning the task, reaching hit rate average values < 0.5 in all test sessions.

Of note, NAC-fed rats displayed significant improvements in hit rate values, which at the sessions performed at 72 h, were significantly higher than those displayed by aged rats not fed with NAC (Figure 5A). To rule out mobility defects in aged rats, the speed at which rats searched for the reward was determined as well. As illustrated in Figure 5B, both young and aged rats, including those fed with NAC, displayed similar speed values when looking for the reward.

Next, the behavior of young and aged rats on a memory task was explored. To achieve this aim, the position of the water-containing reward well was the same in each session. As illustrated in Figure 6A, aged rats displayed hit rate values < 0.5 in all three test sessions. In contrast, both young and NAC-fed aged rats displayed significantly higher hit rate values than aged rats in all three test sessions performed at 24 h, with values close to 1 in the third session. Both young and aged rats, including those fed with NAC, displayed similar speed values when looking for the reward (Figure 6B).

### 2.4. Effects of Training on RyR Channel Responses to Cytoplasmic [Ca^2+^]

As a final step, the effects of training on the frequency of appearance of the three different RyR channel responses to cytoplasmic [Ca^2+^] in hippocampal channels, from young and aged trained rats, was studied (Figure 4, histograms labeled T). In young rats, training induced a significant change (*p* < 0.001) in the emergence of RyR channel responses, reducing the low activity response from 66.7% to 23.8% and increasing the high activity response from 1.7% to 28.6%. In contrast, in aged rats, training did not modify the frequency of appearance of the responses to [Ca^2+^] of RyR channels; in both control and trained rats, the frequency of obtaining low, moderate or high activity responses was around one third (Figure 4).

The administration of NAC to both aged and young rats produced, after training, a similar frequency of appearance of the three responses to cytoplasmic [Ca^2+^] in hippocampal RyR channels (Figure 4, histograms labeled T + NAC).

## 3. Discussion

### 3.1. Summary of Main Findings

In this work, we show, for the first time, the activity of hippocampal RyR channels of aged male rats (22 months old), recorded in vitro, at the single channel level. Hippocampal RyR channels from aged rats showed a twenty-one-fold increase in the frequency of finding RyR channels with the high activity response to [Ca^2+^], and a two-fold reduction in the low activity response, when compared with young rats (3 months old). Aged rats displayed significant defects in learning and memory tests, with hit rate values < 0.5, significantly lower than young rats that exhibited hit rate values near or equal to 1. NAC feeding reversed both age-dependent defects in learning and memory, and produced, in both aged and young rats, a similar frequency pattern of responses to cytoplasmic [Ca^2+^] in RyR channels after training.

### 3.2. Effects of Age on the Responses to [Ca^2+^] of Single RyR Channels

Aging did not modify the three different channel responses to cytoplasmic [Ca^2+^] but significantly increased the probability of the appearance of high activity RyR channels and reduced the probability of the appearance of channels with low activation. This change in channel response may be the result of RyR redox modifications [3,4,5,6,8,9,10,11,12].

The results presented here, which indicate that hippocampal RyR channels from aged rats are more oxidized, agree with previous findings showing higher oxidation levels of the RyR2 channel protein isolated from the hippocampi of aged rats compared to young rats [15]. In addition, the present results indicate that NAC feeding largely reversed RyR channel oxidation state, as evidenced by the decreased frequency of finding channels with the high activity response to [Ca^2+^] displayed by aged rats. These findings complement previous work showing that oral NAC feeding prevents the spatial memory deficits displayed by an Alzheimer’s disease rat model and suppresses the emergence of single RyR channels displaying the high activity response [21]. Hence, these combined results support the beneficial effects of NAC feeding on hippocampal function in age and disease. Accordingly, it is proposed that NAC feeding may be a useful strategy to counteract the increased oxidation of RyR channels that, when reaching high levels of oxidative modifications, would generate unregulated Ca^2+^ signals that presumably have a negative impact on hippocampal learning and memory processes.

Previous reports indicate that ischemia increases the production of ROS and reactive nitrogen species (RNS) [22,23,24,25,26], which participate in normal and pathologic redox signaling by modifying RyR channel activity, among other target proteins that have cysteine residues highly reactive at physiological pH [27]. Of note, hydrogen peroxide promotes the *S*-glutathionylation of RyR channels in hippocampal neurons in primary culture [28] and in skeletal SR vesicles [10]. Moreover, whereas both the RyR2 and RyR3 isoforms present in rat brain cortices exhibit endogenous levels of *S*-glutathionylation and *S*-nitrosylation, ischemia only increases the *S*-glutathionylation and *S*-nitrosylation levels of the RyR2 isoform [22]. In contrast, the *S*-glutathionylation levels of both isoforms increases following incubation of control ER vesicles with NADPH [22], which in the presence of glutathione, sequentially generates superoxide anion and hydrogen peroxide via a sulfenic acid residue intermediate [27]. It was proposed that the intracellular production of ROS/NO is compartmentalized in brain tissue {22], so that they selectively modify the RyR2 and RyR3 isoforms that are differentially distributed in rodent brain cortices [29] and hippocampi [30]. Accordingly, the cellular sources of ROS that are activated by ischemia and aging may target the RyR2 isoform preferentially.

### 3.3. Effects of Age on Spatial Learning and Memory, and Protective Effects of NAC Feeding

Previous studies reported RyR2/RyR3 up-regulation and increased RyR2 oxidation levels in aged rat hippocampi [15]; the resulting anomalous Ca^2+^ release signals may play a significant part in the impaired hippocampal function displayed by aged rodents [16]. The results presented here confirm that aged rats displayed significant defects in the performance of learning and memory tasks in the Oasis maze. Moreover, as mentioned above, direct injections into the hippocampi of amyloid beta peptide oligomers (AβOs), which are causative agents of Alzheimer’s disease, by engaging oxidation-mediated reversible pathways, increase single RyR2 channel activation by Ca^2+^ and cause considerable spatial memory deficits. Previous NAC feeding prevents these noxious effects of AβOs [21]. In agreement, the results presented here show that previous NAC feeding significantly reversed the defective cognitive responses displayed by aged rats. These results indicate that feeding an antioxidant agent such as NAC, a glutathione precursor, may be an eventual protective strategy to counteract the learning and memory defects displayed by aged humans.

### 3.4. Effects of Age on the Responses to [Ca^2+^] of Single RyR Channels from Young and Aged Rats Trained in a Memory Task, and Protective Effects of NAC Feeding

As reported here, hippocampal RyR channels from young rats trained in a spatial memory task displayed a significant increase in the high activity responses to cytoplasmic [Ca^2+^] and a reduction in low activity response. In contrast, RyR channels from trained aged rats did not display significant changes in the distribution of the three responses to Ca^2+^ when compared to untrained aged rats, an indication that training did not increase further RyR channel oxidation in aged rats. Of note, trained aged rats previously fed with NAC displayed a reduction in the frequency of appearance of the high activity response, accompanied by an increase in the frequency of emergence of the low activity response. This frequency distribution is comparable to that displayed by trained and NAC-treated young rats. These novel results suggest that memory training results in increased hippocampal ROS levels, which moderately increase the RyR channel oxidation state, a response that would favor RyR-mediated Ca^2+^-induced Ca^2+^ release, which is required for hippocampal synaptic plasticity, learning and memory processes [2].

## 4. Materials and Methods

### 4.1. Animals

Sprague–Dawley juvenile (7 weeks) and aged (21 months) male rats were procured from the Animal Care Facility of the Faculty of Medicine, Universidad de Chile. Rats were individually housed in a controlled environment with a 12 h light–dark cycle at 21–23 °C, with food and water ad libitum, except when indicated otherwise. All animals were handled daily for 1 week before the beginning of the training sessions. The experimental protocols used in this work complied with the “Guidelines for the Care and Use of Mammals in Neuroscience and Behavioral Research”, The National Academies Press, Washington, DC, USA.

### 4.2. NAC Feeding Protocol

Juvenile and aged rats were fed daily for 21 consecutive days with commercial jelly (1 mL) containing either the antioxidant NAC (200 mg/kg) or vehicle. This oral NAC feeding protocol was maintained during all subsequent procedures, including pre-training and testing in the Oasis maze task.

### 4.3. Spatial Memory Training and Evaluation

The different groups of rats, restricted of water to enhance motivation behavior, were exposed to the spatial memory task in the Oasis maze, a dry-land version of the Morris water maze [31]. All animals were subjected to water restriction for 23 h before the start of each pre-training or training session; water was provided ad libitum for 1 h after these sessions. All rats were pre-trained for 3 d in the Oasis maze, followed by a training period of 6 d. Water-restricted rats were pre-trained during three consecutive daily sessions to familiarize the animal with the testing environment (circular arena provided with visual cues) and the search for the reward (water) in 21 equidistant distributed wells; all wells contained the reward. The ensuing training tasks entailed searching for the reward in one out of 21 wells during six daily sessions. Each session encompassed 10 trials of 1 min duration each, performed at 20–30 s inter-trial intervals. For the learning evaluation the reward was placed in a different well in each session but was kept in the same position during the 10 trials performed in each session. These sessions were conducted at increasing intervals: 24 h between sessions 1 and 2, 48 h between sessions 2 and 3 and 72 h between sessions 3 and 4. During the memory evaluation, the reward was placed in the same well across all sessions and trials. The three sessions were conducted at regular 24 h intervals. Animal behavior was recorded with a video camera in the zenithal position. The position of the animal was monitored continuously during the tests, and the navigation trajectory was reconstructed and analyzed with a customized MATLAB https://www.mathworks.com/products/matlab.html?s_tid=hp_products_matlab, accessed on 3 February 2025 (MathWorks) routine. One hour after the end of the sixth session, animals underwent euthanasia by decapitation, and the hippocampus was collected for determination of RyR single channel activity.

### 4.4. Statistical Analysis

Results are expressed as the mean ± SE. Statistical analysis between groups was performed with two-way ANOVA, one-way ANOVA followed by Holm–Sidak post hoc test, as indicated in the respective figure legends. Comparison between two groups was performed by two-tailed Student’s *t*-test. All statistical analyses were performed using Sigma Plot version 12.0.

### 4.5. Determination of RyR Single Channel Activity

The rats were decapitated with guillotine and their brains quickly removed. Brain cortices and the whole hippocampus were dissected as described [13,14,18]. Cleaned dissected tissue was homogenized in sucrose buffer with protease inhibitors (0.3 mM sucrose, 20 mM MOPS Tris, pH 7.0, 0.4 mM benzamidine, 10 mg/mL trypsin inhibitor; 10 mg/mL pepstatin) and centrifuged at 5000× *g* for a duration of 20 min. The supernatants were centrifuged at 100,000× *g* for 1 h; the resulting pellets were resuspended in sucrose buffer with protease inhibitors and frozen in aliquots at −80 °C.

During channel recording at 22 ± 2 °C, the cis-(cytoplasmic) solution contained 0.5 mM Ca^2+^-HEPES, and 225 mM HEPES–Tris, pH 7.4, while the trans (intra-reticular) solution contained 40 mM Ca^2+^-HEPES, 15 mM Tris-HEPES, pH 7.4. The lipid bilayer was held at 0 mV. Current data were filtered at 400 Hz (−3 dB) using an eight-pole low-pass Bessel type filter (902B; Frequency Devices, Haverhill, MA, USA) and digitized at 2 kHz with a 12-bit analog-to-digital converter (LabMaster DMA Interface; Scientific Solutions; https://scientific-solutions.com/products/labmaster_dma/labmaster_dma_index.html, accessed on 15 September 2024) using the AxoTape 8 (Molecular Devices) commercial software. P_o_ values were computed using the pClamp 6 (Molecular Devices) commercial software. RyR channels were classified as low (P_o_ < 0.1 at all [Ca^2+^] tested), moderate (P_o_ > 0.1 in the [Ca^2+^] range of 10–100 µM, with reduction in P_o_ at 500 µM [Ca^2+^]) or high (P_o_ near 1.0 in the range of 3–500 µM [Ca^2+^]) activity channels [7,9,11]. Data are expressed as the mean ± SE. Comparison between frequency histograms was performed by the Chi-Squared test.

The Ca^2+^-dependence of single RyR channel P_o_ values was fitted with the following general function [22]:P_o_ = P_o max_ × [Ca^2+^]^n^/([Ca^2+^]^n^ + K_a_^n^) × K_i_/([Ca^2+^] + K_i_)(1)

Equation (1) gives P_o_ values as a function of cis-[Ca^2+^]. P_o max_ corresponds to the theoretical P_o_ value of maximal activation by cis-[Ca^2+^]. K_a_ and K_i_ correspond to the Ca^2+^ concentrations for half-maximal activation and inhibition, respectively, of the channel activity, and n is the Hill coefficient for Ca^2+^ activation.

## Figures and Tables

**Figure 1 ijms-26-02101-f001:**
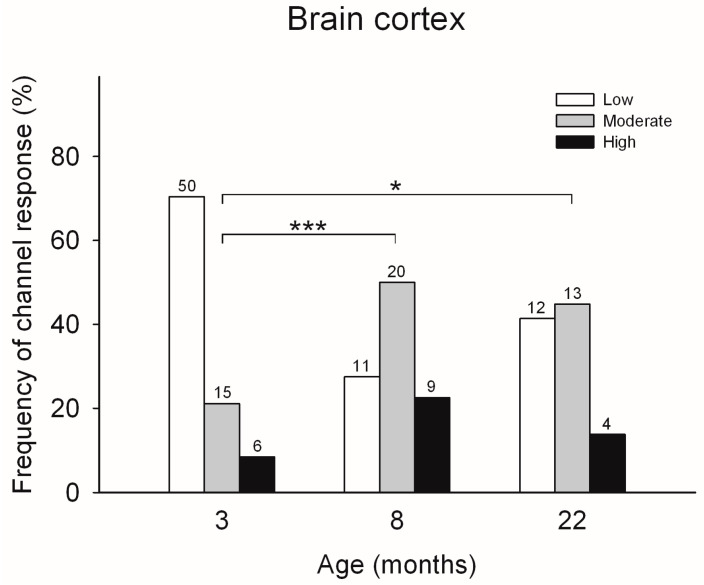
Frequency of incorporation of RyR channels isolated from the brain cortices of rats with different ages. All channels displayed either the low, moderate or high activity response. On top of each bar is the number (*n*) of channels displaying either response to cytoplasmic [Ca^2+^], recorded after fusion of the ER vesicle preparation. The frequencies of emergence of the three types of responses differed between channels from the brain cortex of rats at 3, 8 or 22 months of age (***: *p*< 0.001; *: *p* < 0.05; Chi-squared test).

**Figure 2 ijms-26-02101-f002:**
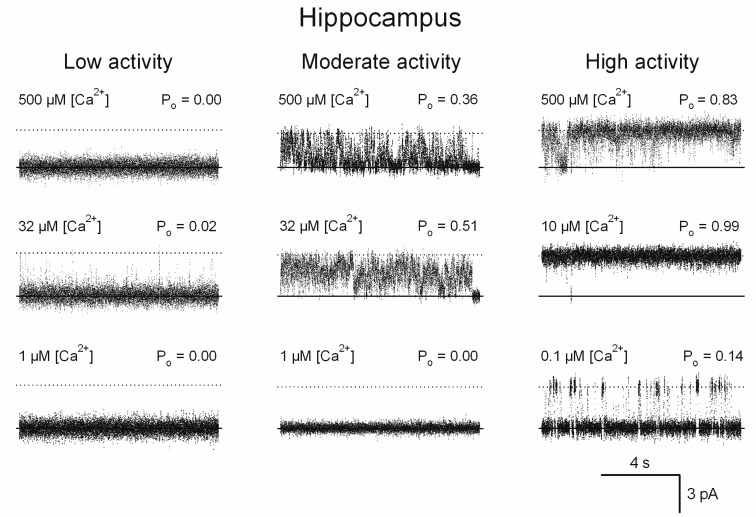
Representative current recordings of hippocampal single RyR channels that displayed low, moderate or high activity responses to cytoplasmic [Ca^2+^]. [Ca^2+^] is given at the upper left of each trace. Average P_o_ values, calculated from at least 50 s of continuous recordings, are given at the top right of each trace. The lipid bilayer was held at 0 mV. Channels open upwards.

**Figure 3 ijms-26-02101-f003:**
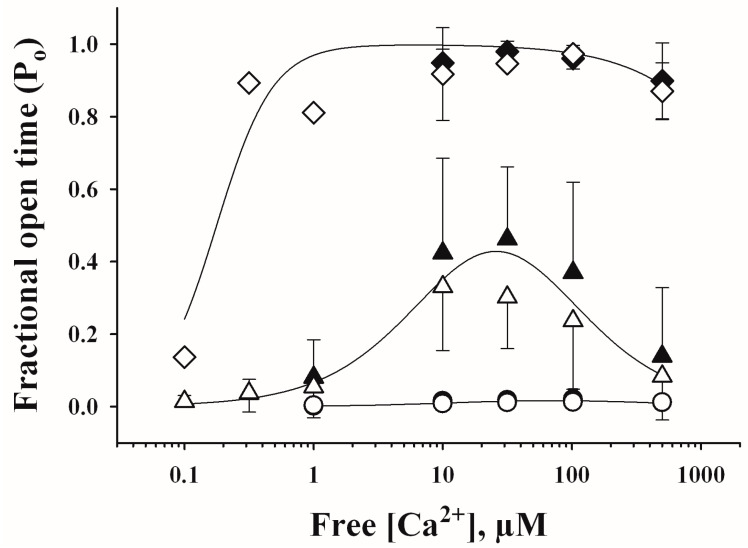
Responses to cytoplasmic [Ca^2+^] displayed by hippocampal RyR channels from young (open symbols) and aged (filled symbols) rats. Fractional open times (P_o_) of low (circles), moderate (triangles) and high activity (diamonds) channels are depicted as a function of [Ca^2+^]. Symbols and error bars depict the mean and SE values, respectively. Continuous lines represent the best nonlinear fits to the Equation (1) (see Materials and Methods) of all individual experimental data values obtained with single RyR channels that displayed either low, moderate or high activity. Nonlinear fitting parameter values are displayed in Table 1.

**Figure 4 ijms-26-02101-f004:**
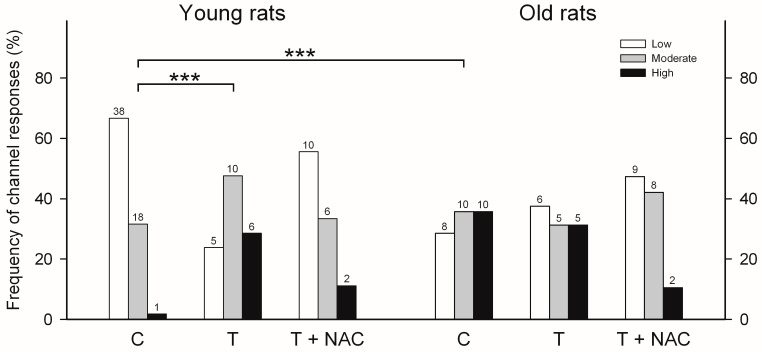
Frequency of incorporation of RyR channels with low, moderate or high activity isolated from the hippocampi of young (aged 3 months) or aged rats (aged 22 months). Depicted on each bar is the number (*n*) of channels displaying either response to cytoplasmic [Ca^2+^] recorded after fusion of ER vesicle preparations. Histograms for control (C), trained (T) and trained plus *N*-acetylcysteine (T + NAC) are displayed for young and old rats. The significant differences between the histograms are indicated (***: *p* < 0.001).

**Figure 5 ijms-26-02101-f005:**
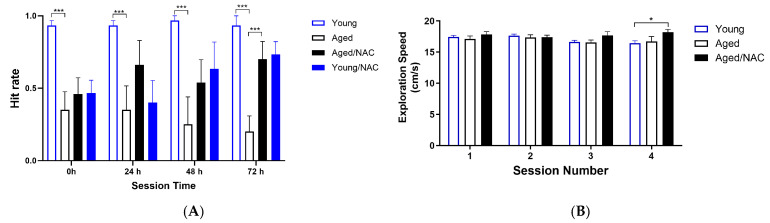
Hit rate (**A**) and exploration speed (**B**) during the learning task displayed by young and aged rats. Values represent the mean ± SE (*n* = 4). Sessions were performed with increasing time intervals; session 2 was performed 24 h after session 1; sessions 3 and 4 were performed 48 h and 72 h after session 1, respectively. Statistical analysis was performed using two-way ANOVA followed by the Holm–Sidak post hoc test. Significant differences are indicated as follows: * *p* < 0.05; *** *p* < 0.001, comparing young with aged rats fed with NAC.

**Figure 6 ijms-26-02101-f006:**
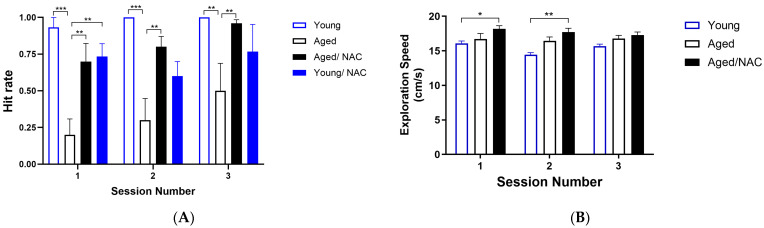
Hit rate (**A**) and exploration speed (**B**) during the memory task displayed by young and aged rats. Sessions were conducted at regular 24 h intervals. Values represent the mean ± SE (*n* = 4). Statistical analysis was performed using two-way ANOVA followed by the Holm–Sidak post hoc test. Significant differences are indicated as follows: *: *p* < 0.05; **: *p* < 0.01; ***: *p* < 0.001, comparing young with aged rats fed with NAC.

**Table 1 ijms-26-02101-t001:** Fitting parameters for the three RyR channel responses.

Calcium Response	K_a_ (µM)	n_Hill_	K_i_ (µM)	P_o max_
Low	374 ± 107 ^a^	1 ^b^	13 ± 4 ^a^	0.65 ^b^
Moderate	14 ± 2 ^a^	1 ^b^	49 ± 7 ^a^	1.00 ^b^
High	0.18 ± 0.03 ^a^	2 ^b^	3902 ± 767 ^a^	1.00 ^b^

Parameter ± SE values were obtained by fitting to Equation (1) (see Materials and Methods) all individual data obtained with single channels, from the hippocampus of young or aged rats, that displayed low, moderate or high activity. ^a^ Significant differences (*p* < 0.001) between corresponding parameter values were found. Parameter values of moderate activity channels were compared with the respective values of low activity channels. Values of high activity channels were compared with the respective values of moderate activity channels. ^b^ Parameter was fixed to the indicated value for data fitting [20].

## Data Availability

Data contained within the article.

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
