# Peer review of "Aging Favors Calcium Activation of Ryanodine Receptor Channels from Brain Cortices and Hippocampi and Hinders Learning and Memory in Male Rats"

_ijms, 2025, doi:10.3390/ijms26052101_

Round 1

Reviewer 1 Report

Comments and Suggestions for Authors

The authors investigate the effects of aging on ryanodine receptor (RyR) calcium channels in the brain and their role in learning and memory impairment in male rats. The manuscript highlights that aging induces oxidative modifications in RyR channels, shifting their activity from a low to a high calcium response state, which disrupts calcium homeostasis. Behavioral experiments using the Oasis maze demonstrated significant deficits in spatial learning and memory in aged rats compared to young ones. The study also explores the therapeutic potential of N-acetylcysteine (NAC), an antioxidant, which reversed age-related RyR oxidation and restored cognitive performance. Training-induced changes in RyR activity were observed in young but not aged rats, further emphasizing the impact of oxidative stress on hippocampal function. The findings suggest that RyR oxidation is a critical factor in age-related cognitive decline and propose antioxidant therapy as a potential intervention.

While the manuscript is well-structured and written, I suggest the following adjustments to enhance its clarity and depth:

1.      Introduction: Please expand the introduction to include more information about the known oxidative post-translational modifications of RyR and their impact on channel activity. Specifically, discuss which post-translational modification sites are affected and what is known about their roles in aging and aging-associated conditions.

2.      Figure 3: The use of white and black symbols in Figure 3 makes it difficult to distinguish the data points. Would it be possible to split the figure into three separate panels for clarity and better visualization?

3.      NAC Treatment and Post-Translational Modifications: While the manuscript demonstrates that NAC treatment modifies the responsiveness of RyR, it remains unclear whether this effect is directly associated with post-translational modifications of the calcium channel. Could the authors explore or discuss the characterization of these modifications in association with NAC treatment? Insights from cell culture experiments or relevant published studies could strengthen this section.

Author Response

  1. Introduction: Please expand the introduction to include more information about the known oxidative post-translational modifications of RyR and their impact on channel activity. Specifically, discuss which post-translational modification sites are affected and what is known about their roles in aging and aging-associated conditions.

    Answer.  Thanks for your comment. The Introduction section was expanded (pages 1 and 2) to include more information and references about the known oxidative post-translational modifications of RyR channels and their impact on channel activity, including additional details on their post-translational modification sites, and their roles in aging and aging-associated conditions.

  2. Figure 3: The use of white and black symbols in Figure 3 makes it difficult to distinguish the data points. Would it be possible to split the figure into three separate panels for clarity and better visualization?

    Answer.  To clarify the results presented in Figure 3, the size of Figure 3 was magnified twice.

  3. NAC Treatment and Post-Translational Modifications: While the manuscript demonstrates that NAC treatment modifies the responsiveness of RyR, it remains unclear whether this effect is directly associated with post-translational modifications of the calcium channel. Could the authors explore or discuss the characterization of these modifications in association with NAC treatment? Insights from cell culture experiments or relevant published studies could strengthen this section.

    Answer.  We discuss now in more detail (pages 7 and 8) the effects of NAC treatment on RyR channel redox modifications and activity.

Reviewer 2 Report

Comments and Suggestions for Authors

I reviewed the manuscript entitled Aging favors calcium activation of ryanodine receptor (RyR) channels from brain cortex and hippocampus and hinders learning and memory in male rats.

I agree to accept this manuscript after major revision. 

1) Abbreviations and periods in titles are not recommended. It is suggested to change to Aging favors calcium activation of ryanodine receptor channels from brain cortex and hippocampus and hinders learning and memory in male rats

2) It is more accurate to change calcium concentration to calcium ion concentration. Please check and revise the full text.

3)  Keywords: The initial of spatial should be capitalized.

4) S-nitrosylation and/or S-glutathionylation of SH residues of skeletal or cardiac RyR channels enhance Ca2+-induced Ca2+ release (CICR), I found that CICR only appears twice in the entire text. It is necessary to use abbreviations only if they appear three times or more, as excessive abbreviations can confuse readers. Please follow this principle to check and modify the entire text.

5) N-acetylcysteine, N should be italicized.

6) 2.1. Effects of age on the Ca2+-induced responses of RyR channels from rat brain cortex. The first letter of each actual word in the secondary title needs to be capitalized. Except for 2.1, other secondary titles also need to be modified.

7) Try to minimize the use of 'we' or 'our' in scientific papers and ensure rigor and objectivity.

8) Figure 1. When it comes to statistics, p should be italicized.

9) causes single RyR channel from rat brain cortex to display higher degrees of activation in response to Ca2+ (8,9-11). (8,9-11) should change to [8,9-11].

10) Based on these findings, it is proposed that NAC feeding may be a useful strategy to counteract the increased RyR channel oxidation, which when reaching high levels presumably impacts negatively on hippocampal learning and memory processes. Does' which 'refer to NAC? If so, the latter half of the sentence needs to be supported by references.

11) Juvenile and aged rats were fed daily for 21 consecutive days with commercial jelly (1 ml), 1 ml should change to 1 mL.

12) All rats were pre-trained for 3 days in the Oasis maze, followed by a training period of 6 days. 3 days should change to 3 d. 6 days should change to 6 d. Use international units instead of words.

13) 10 mg/ml trypsin inhibitor; 10 mg/ml pepstatin, mg/mL is right. frozen in aliquots at -80 °C. -80 °C should change to  -80°C, There should be spaces between numbers and units, except for °C and %. Check and revise the entire text.

14) I have read all the references and found some issues. Ref 2, the volume needs to be italicized. Ref 3, 499 –509, Delete spaces after 499. Ref 24, Biol Res should be italicized. Ref 25, 633,96-103.  633 needs to be italicized, there should be a space after the comma. Ref 27, Antioxid Redox Signal should be italicized. Ref 28 is the same issue. Meanwhile, Mg2+, 2+ should be superscripted.

15) The present study reveals that the response of ryanodine receptor (RyR) channels to cytoplasmic calcium concentration ([Ca2+]) is modulated by redox signaling. It demonstrates that RyR channels exhibit varying activity responses—low, moderate, and high—depending on their oxidation state. In aged rats, the study finds a shift in RyR channels from the brain cortex or hippocampus towards higher activity responses, which favors Ca2+-induced Ca2+ release. Furthermore, the study shows that aged rats exhibit learning and memory defects when tested in the Oasis maze. Notably, prior oral administration of N-acetylcysteine prevented both the age-related effects on RyR channel activation and the learning and memory impairments observed. This study proposes that redox-sensitive neuronal RyR channels play a role in the learning and memory disruptions displayed by aged rats.

16) An excessively high repetition rate (45%) is unacceptable, and the author must make revisions to prove that there is no possibility of plagiarism or self plagiarism.

17) The biggest problem with this study is many details need to be modified and improved. Too many issues can make people feel that the author's attitude is not rigorous.The author must take them seriously and make necessary revisions.

18) The conclusion is consistent with the evidence and arguments provided. All the main questions raised by the author have been resolved.

Author Response

1) Abbreviations and periods in titles are not recommended. It is suggested to change to Aging favors calcium activation of ryanodine receptor channels from brain cortex and hippocampus and hinders learning and memory in male rats

Answer:  Thanks for this and the following comments. The title was modified as suggested.

2) It is more accurate to change calcium concentration to calcium ion concentration. Please check and revise the full text.

Answer:  We changed “calcium concentration” to “calcium ion concentration” and revised the full text accordingly.

3)  Keywords: The initial of spatial should be capitalized.

Answer:  We corrected the text as suggested.

4) S-nitrosylation and/or S-glutathionylation of SH residues of skeletal or cardiac RyR channels enhance Ca2+-induced Ca2+ release (CICR), I found that CICR only appears twice in the entire text. It is necessary to use abbreviations only if they appear three times or more, as excessive abbreviations can confuse readers. Please follow this principle to check and modify the entire text.

Answer: We suppressed this definition and corrected the text accordingly.

5) N-acetylcysteine, N should be italicized.

Answer: We italicized “N-acetylcysteine” thorough all the text.

6) 2.1. Effects of age on the Ca2+-induced responses of RyR channels from rat brain cortex. The first letter of each actual word in the secondary title needs to be capitalized. Except for 2.1, other secondary titles also need to be modified.

Answer:  We corrected the text accordingly.

7) Try to minimize the use of 'we' or 'our' in scientific papers and ensure rigor and objectivity.

Answer:  We made the changes accordingly throughout all the text.

8) Figure 1. When it comes to statistics, p should be italicized.

Answer:  We italicized all corresponding statistical “p”.

9) causes single RyR channel from rat brain cortex to display higher degrees of activation in response to Ca2+ (8,9-11). (8,9-11) should change to [8,9-11].

Answer: We changed the brackets, now cited with different numbers.

10) Based on these findings, it is proposed that NAC feeding may be a useful strategy to counteract the increased RyR channel oxidation, which when reaching high levels presumably impacts negatively on hippocampal learning and memory processes. Does' which 'refer to NAC? If so, the latter half of the sentence needs to be supported by references.

Answer: The word “which” on page 7 does not refer to NAC. Moreover, in the revised version we substantially modified this part of  the text.

11) Juvenile and aged rats were fed daily for 21 consecutive days with commercial jelly (1 ml), 1 ml should change to 1 mL.

Answer: We changed “ml” to “mL”.

12) All rats were pre-trained for 3 days in the Oasis maze, followed by a training period of 6 days. 3 days should change to 3 d. 6 days should change to 6 d. Use international units instead of words.

Answer: We changed “days” to “d”.

13) 10 mg/ml trypsin inhibitor; 10 mg/ml pepstatin, mg/mL is right. frozen in aliquots at -80 °C. -80 °C should change to  -80°C, There should be spaces between numbers and units, except for °C and %. Check and revise the entire text.

Answer:  We made the changes accordingly throughout the text.

14) I have read all the references and found some issues. Ref 2, the volume needs to be italicized. Ref 3, 499 –509, Delete spaces after 499. Ref 24, Biol Res should be italicized. Ref 25, 633,96-103.  633 needs to be italicized, there should be a space after the comma. Ref 27, Antioxid Redox Signal should be italicized. Ref 28 is the same issue. Meanwhile, Mg2+, 2+ should be superscripted.

Answer:  We thank you again for your review of our manuscript. We made all corrections to the references. 4 references were changed.

15) The present study reveals that the response of ryanodine receptor (RyR) channels to cytoplasmic calcium concentration ([Ca2+]) is modulated by redox signaling. It demonstrates that RyR channels exhibit varying activity responses—low, moderate, and high—depending on their oxidation state. In aged rats, the study finds a shift in RyR channels from the brain cortex or hippocampus towards higher activity responses, which favors Ca2+-induced Ca2+ release. Furthermore, the study shows that aged rats exhibit learning and memory defects when tested in the Oasis maze. Notably, prior oral administration of N-acetylcysteine prevented both the age-related effects on RyR channel activation and the learning and memory impairments observed. This study proposes that redox-sensitive neuronal RyR channels play a role in the learning and memory disruptions displayed by aged rats.

Answer:  We thank you.

16) An excessively high repetition rate (45%) is unacceptable, and the author must make revisions to prove that there is no possibility of plagiarism or self plagiarism.

Answer:  Thanks for all your careful revision of our work. The text was thoroughly modified to avoid unwanted plagiarism. 

17) The biggest problem with this study is many details need to be modified and improved. Too many issues can make people feel that the author's attitude is not rigorous.The author must take them seriously and make necessary revisions.

Answer.  Thanks for all your careful revision of our work. The text was thoroughly modified to comply with your instructions.

18) The conclusion is consistent with the evidence and arguments provided. All the main questions raised by the author have been resolved.

Answer.  Thank you

Round 2

Reviewer 1 Report

Comments and Suggestions for Authors

Thank you for addressing my previous comments.

I recommend reviewing the consistency of your writing, particularly in how you format p-values—sometimes they include separators, and other times they do not.

Other than that, I have no further suggestions.

Author Response

Reviewer 1

Thank you for addressing my previous comments.

Answer: You are welcome.

I recommend reviewing the consistency of your writing, particularly in how you format p-values—sometimes they include separators, and other times they do not.

Answer: Thank you for your suggestion. We revised all the text and corrected 2 format inconsistencies found. We highlighted all corrections made.

Other than that, I have no further suggestions.

Answer: Thank you for your revision.

Reviewer 2 Report

Comments and Suggestions for Authors

I agreed to accept this manuscript in this form.

Author Response

I agreed to accept this manuscript in this form.

Answer: Thank you for your kindness.